# Intake of Boiled Potato in Relation to Cardiovascular Disease Risk Factors in a Large Norwegian Cohort: The HUNT Study

**DOI:** 10.3390/nu12010073

**Published:** 2019-12-27

**Authors:** Trine Moholdt, Brooke L. Devlin, Tom Ivar Lund Nilsen

**Affiliations:** 1Department of Circulation and Medical Imaging, Norwegian University of Science and Technology, 7491 Trondheim, Norway; 2Women’s Clinic, St Olavs Hospital, Trondheim University Hospital, 7030 Trondheim, Norway; 3Department of Dietetics, Nutrition and Sport, La Trobe University, Melbourne 3086, Victoria, Australia; 4Department of Public Health and Nursing, Norwegian University of Science and Technology, 7491 Trondheim, Norway; tom.nilsen@ntnu.no; 5Clinic of Anaesthesia and Intensive Care, St Olavs Hospital, Trondheim University Hospital, 7030 Trondheim, Norway

**Keywords:** diet, blood pressure, body mass index, cholesterol, sex

## Abstract

Overall potato consumption is positively associated with cardiovascular disease (CVD) risk factors, such as measures of adiposity. However, few studies have explicitly stated the preparation method of potatoes, which may impact these associations. We examined cross-sectional associations between self-reported dietary intake of boiled potatoes and levels of body mass index, waist circumference, blood pressure, and blood lipids among 43,683 participants in the HUNT Study, Norway in 2006–2008. All estimated associations were adjusted for possible imbalance in age, sex, physical activity, smoking, intake of other foods and alcohol between categories of boiled potato consumption. Overall, there were no large differences in mean levels of CVD risk factors between categories of boiled potato consumption. Compared to the reference group of individuals who consumed boiled potatoes less than once/week, those who reported eating boiled potatoes every day had slightly higher prevalence of high waist circumference (odds ratio [OR] 1.16, 95% confidence interval [CI] 1.05–1.29), high triglycerides levels (OR 1.20, 95% CI 1.07–1.34), and metabolic syndrome (OR 1.17, 95% CI 1.03–1.33). In summary, consumption of boiled potatoes showed weak and small associations with the CVD risk factors under study, but the cross-sectional design prevents us from drawing any firm conclusions.

## 1. Introduction

Potato is the fourth most-produced crop in the world and the second most consumed food in the US [1]. Although potatoes provide several key nutrients including potassium, dietary fibre and vitamin C and are low in sodium, they have been disparaged for their high carbohydrate content [2] and high glycaemic index. A systematic review on the association between dietary potato intake and obesity, metabolic syndrome, and cardiovascular disease (CVD) concluded that there is not convincing evidence to suggest an association between intake of potatoes and risks of obesity, type 2 diabetes, or CVD [3]. Overall potato consumption has been positively associated with measures of adiposity, such as body mass index (BMI), waist circumference and weight gain [3]. Although intake of french fries has been reported to have a stronger positive association with measures of adiposity than intake of boiled, baked or mashed potatoes [4,5], few studies have information on preparation method (e.g., boiled, baked, mashed or fried) [3]. 

Studies have also reported differential associations in men and women [3]. Two large cohort studies of US women observed a positive association between potato consumption and risk of T2D [6,7], whereas others found no association between potato intake and T2D risk in men [8]. Although a high intake of potatoes has been associated with lower increase in waist circumference among men [9] and a higher increase among women [10], differential associations for other CVD risk factors have not been examined. Furthermore, there is evidence suggesting that the association between potato consumption and risk of T2D is stronger in individuals with obesity compared to in those who are lean [6,11]. Whether this is reflected in the prevalence of other CVD risk factors is not known, but it is conceivable that both potato intake and CVD risk factor levels are influenced by body mass index (BMI).

The aim of the present study was to describe possible cross-sectional associations between habitual intake of boiled potatoes and CVD risk factors, including BMI, waist circumference, blood pressure and blood lipids, in a large Norwegian population. We also compared these associations between men and women, as well as between individuals within different BMI categories (normal weight versus overweight/obese). 

## 2. Methods

### 2.1. Study Population

The HUNT3 survey was conducted in 2006–2008 and includes a total population-based cohort from one county in Norway. Every citizen aged 20 years and older was invited to participate in the survey, with 50 807 (54% of those invited) participating (59% women). HUNT3 included several study parts, from completing extensive questionnaires to a number of additional clinical tests (including blood sampling and measurements of blood pressure) and interviews. Detailed information about the HUNT-studies and cohort profile is available at [12]. In this project, we included participants without known CVD from HUNT3 with valid data on potato consumption (*n* = 43,683).

### 2.2. Assessment of Boiled Potato Intake, Co-Variates, and Outcomes

The participants answered questions about their habitual intake of various food groups and foods, such as vegetable, fruits, fish, processed meat, pasta and rice. The dietary questions in HUNT3 were selected from a questionnaire used in the Oslo Health Study of 2001 and were validated against intake of food and food groups using a 14 day diet diary [13]. Our primary exposure variable was the frequency of boiled potato intake, as assessed by the question “How often do you normally eat boiled potatoes?” We categorised boiled potato intake into four frequencies; (1) <1 per week, (2) 1–3 per week, (3) 4–6 per week, and (4) ≥1 per day. 

Blood pressure was measured using a Dinamap 845XT (Critikon) based on oscillometry. Blood pressure was measured three times, with one-minute intervals between the measurements. In the analysis, we used the mean of the second and third measurement. We defined hypertension as systolic blood pressure ≥130 mmHg and/or diastolic blood pressure ≥85 mmHg or currently taking blood pressure medication. Height and weight were measured with the participants wearing light clothes and no shoes, height to the nearest centimetre and weight to the nearest half kilogram. Body mass index was calculated as kg/m^2^. Waist circumference was measured with a measuring tape at the level of the umbilicus, to the nearest centimetre with the participant standing with arms hanging relaxed. We defined high waist circumference as >88 cm for women and >102 cm for men.

Blood was sampled via forearm venepuncture by certified personnel at HUNT health examination stations. Serum cholesterol was analysed by enzymatic cholesterol esterase methodology, high-density lipoprotein cholesterol (HDL) by Accelerator selective detergent methodology, and triglycerides by Glycerol Phosphate Oxidase methodology (all from Abbot, Clinical Chemistry). The coefficients of variation of the assays employed were 1.4%–1.7% for total cholesterol, 3.0%–4.4% for HDL, and 2.5%–4.5% for triglycerides. We defined high total cholesterol as ≥6 mmol/L, high triglycerides as ≥1.7 mmol/L and low HDL as <1.30 mmol/L for women and <1.04 mmol/L for men.

Participants self-reported their smoking status (categorised as never, former, or current smoking) and their physical activity level (categorised as inactive, <1 weekly bout of exercise, 2–3 weekly bouts of exercise, or four or more weekly bouts of exercise).

Metabolic syndrome was defined according to modified ATPIII criteria [14] requiring three out of the four following factors to be present based on the above cut-offs: high waist circumference, hypertension, high triglycerides and low HDL cholesterol. Fasting glucose was not measured in the HUNT study and could not contribute to the definition of metabolic syndrome.

### 2.3. Statistical Analysis

Mean differences in cardiovascular risk factor levels between categories of potato consumption were estimated from linear regression, whereas logistic regression was used to estimate odds ratios (ORs) for an adverse risk factor level between the groups. Participants who reported boiled potato consumption less than once per week were used as reference category for all comparisons. We adjusted for the following factors that we considered could be related both to potato consumption and the metabolic factors; sex (woman, man), age (years), occupation (mostly sedentary, much walking, much lifting, heavy physical work), frequency of physical activity per week (none, <1, 1, 2–3, ≥4 days), smoking (never, former, current), alcohol consumption (≤1 per month, ≤1 per week, ≥2 per week, abstainer), and weakly consumption of vegetables (≤3 times per week, 4–6 times per week, ≥1 times per day), pasta/rice (≤3 times per month, 1–3 times per week, ≥4 times per week), fish (≤3 times per month, 1–3 times per week, ≥4 times per week) and processed meat (≤3 times per month, 1–3 times per week, ≥4 times per week). Analyses of blood pressure and blood lipids were also adjusted for BMI (kg/m^2^). We also conducted analyses separately for women and men, as well as for those with BMI ± 25 kg/m^2^, and we assessed possible statistical interaction in a likelihood ratio test of a product term between these factors and potato consumption Finally, the main analyses were repeated in a sensitivity analysis excluding people who reported a family history of myocardial infarction (i.e., myocardial infarction before 60 years of age in a first-degree relative). The precision of the estimated associations was assessed by a 95% confidence interval (CI). All analyses were conducted using Stata © 1985–2017 StataCorp LLC (College Station, TX, USA).

## 3. Results

Table 1 shows selected characteristics of the participants, according to boiled potato consumption.

The intake of boiled potatoes was high in this population, with 67% eating boiled potatoes four times or more per week. Apart from increasing age with increasing consumption of boiled potato, the characteristics of participants were similar between the categories of potato consumption.

Table 2 shows the associations between boiled potato intake and cardiometabolic risk factors.

### 3.1. Overweight/Obesity

People who consumed boiled potatoes more than four times per week had a slightly higher mean BMI and waist circumference (Table 2). The prevalence of BMI ≥25 kg/m^2^ and high waist circumference was somewhat higher in those who reported to consume boiled potatoes more than once per week, compared with those eating boiled potatoes less than once per week (Figure 1, Appendix A). 

Analyses stratified by sex showed that the positive association between boiled potato consumption and mean BMI and waist circumference was largely confined to women (Table 3 and Appendix A), with *p*-values from tests of interaction between sex and potato consumption of <0.01 and 0.01, respectively (Table 3). The results from analyses stratified by BMI ± 25 kg/m^2^ were largely similar between strata, although the positive association between boiled potato consumption and waist circumference was somewhat stronger in people with BMI <25 kg/m^2^ than in those with the highest BMI (Table 4).

### 3.2. Blood Pressure

We found no clear association between boiled potato consumption and mean blood pressure (Table 2) or prevalence of hypertension (Figure 1). Results from analyses stratified by sex- and BMI were in line with this (Table 3 and Table 4, Appendix A). 

### 3.3. Blood Lipids

There were negligible associations between boiled potato consumption and mean total cholesterol and triglycerides, whereas boiled potato consumption was associated with slightly lower HDL cholesterol (Table 2). There was a slightly higher prevalence of people with high total cholesterol (≥6 mmol/L), high triglycerides (≥1.7 mmol/L) and low HDL (<1.30 mmol/L for women, and <1.04 mmol/L for men) among those eating most boiled potatoes compared to the reference group (Figure 1 and Appendix A). Sex specific analyses showed somewhat stronger associations with HDL cholesterol in women than in men (Table 3 and Appendix A), and analyses stratified by BMI ± 25 kg/m^2^ showed somewhat stronger associations with the prevalence of low HDL in those with the lowest compared to the highest BMI (Appendix A). 

### 3.4. Metabolic Syndrome

There was a slightly higher prevalence of metabolic syndrome in people who consumed boiled potatoes daily compared to the reference group (OR 1.17, 95% CI 1.03–1.33) (Appendix A). This association was somewhat stronger in women than in men (Appendix A), and in those with BMI <25 kg/m^2^ compared to those who were overweight/obese (Appendix A). 

### 3.5. Sensitivity Analysis

We repeated the main analysis excluding individuals with a family history of myocardial infarction, since it is conceivable that a family history could influence dietary habits. However, the results remained largely similar to those in the main analysis (data not shown). 

## 4. Discussion

We investigated the cross-sectional associations between intake of boiled potatoes and CVD risk factors in a large Norwegian population where boiled potatoes constitute an important part of the diet. Two-thirds of the population reported to eat boiled potatoes four times or more per week. Overall, we found that intake of boiled potatoes showed no or weak positive associations with adverse levels of the CVD risk factors under study. 

We observed a somewhat stronger association between intake of boiled potatoes and measures of adiposity (both BMI and waist circumference) in women than in men. Although our results are cross-sectional in nature, they are in line with studies from Halkjær and colleagues who observed that a high intake of potatoes seemed to prevent an increase in waist circumference for men [9], whereas the opposite was true for women [10]. A study from Iran, where they typically consume boiled potatoes, however, reported that both men and women who consumed potatoes more than once per week had a lower BMI than those with less frequent potato consumption [15]. Although potatoes are classified as a vegetable with known nutritional benefits, potatoes contain large amounts of carbohydrate and have a high glycaemic index, resulting in a rapid rise in blood glucose [16]. Foods with a high glycaemic index have been proposed to promote weight gain by redirecting nutrients from oxidation in insulin-sensitive tissues (such as skeletal muscle) to storage as fat [17]. 

High intake of boiled potatoes was weakly associated with an adverse lipid profile. We observed minor sex-differences also for lipids according to boiled potato consumption. Only men with higher intake of boiled potatoes had elevated levels of triglycerides and total cholesterol, whereas in both sexes, potato consumption was associated with lower levels of HDL cholesterol. There are limited studies in humans on the effect of potato consumption on lipid levels. Khosravi-Boroujeni and co-workers reported no associations between potato consumption and blood lipids among Iranian men or women [15]. In a cross-over design, ingestion of potato fibres decreased postprandial plasma levels of total and esterified cholesterol, but had no effect on fasting levels, in healthy males and females [18]. Also in males and females with the metabolic syndrome, fasting blood lipids remained unchanged after 12 weeks of consumption of a diet composed of oat, wheat bread and potatoes [19]. However, in that study, cholesterol synthesis was decreased, and cholesterol absorption increased, compared to baseline and to a group who ate a diet composed of rye bread and pasta. The latter two studies did not, however, undertake sex-stratified analyses. 

There were no associations between potato intake and blood pressure or hypertension in our study. Primary hypertension (also known as idiopathic or essential hypertension) is typically seen in Western societies that consume processed food high in sodium and low in potassium, causing a high sodium-to-potassium ratio [20]. Potatoes contain low amount of sodium (4 mg/100 g) and are rich in potassium and vitamin C (500 mg/100 g, equal to almost 20% of the recommended daily intake). Studies reporting the relationship between consumption of potatoes and hypertension have given mixed results [21,22,23]. Data from the China Health and Nutrition Survey 1989–2011 showed that total potato consumption was prospectively associated with hypertension [23]. In China, potatoes are most often prepared by stir-frying and therefore introducing additional intake of fat and sodium. Worth noticing in that study, when non-potato consumers were excluded from the analysis, higher total potato consumption and stir-fried potato consumption were associated with lower risk of hypertension [23]. Using data from the Nurses Health Study (women) and the Health Professionals Follow-up Study (men), Borgi et al. [21] reported independent prospective associations of higher consumption of potatoes (including baked, boiled, mashed and french fries) with an increased risk of hypertension. Conversely, data from two Spanish cohort studies showed no association between potato consumption and changes in blood pressure over 4–6.7 years or with the risk of hypertension in older adults [22]. 

Strengths of the current study include the population-based nature of the data, detailed information on lifestyle and health related factors, and a large sample size providing precise estimates of association. Although there was some evidence of differential associations between men and women, and between people with high and low BMI, it should be noted that the large number of participants may result in small p-values from interaction tests despite relatively small differences in the observed associations. 

There are some limitations to our study. The HUNT study did not collect information on total energy intake of participants. Therefore, it is not possible to determine if individuals with high intake of potato also overall consume more total energy. As such, it can be assumed that the potato consumption could merely reflect total energy intake. Furthermore, the questions within the HUNT study about potato consumption did not capture information about the portion sizes of boiled potato consumption, only the frequency of intake. There is no information available about consumption of potatoes prepared in other ways (i.e., fried) and no information about whether the boiled potatoes were consumed with or without the skin. Even if we cannot exclude that the response rate of 54% affected the results, it has been argued that representativeness is not a prerequisite for generalization [24]. These limitations along with the fact the study was cross-sectional in design, need to be considered when interpreting the study results. According to a dose–response meta-analysis of cohort studies, consumption of potatoes is strongly associated with increased risk of type 2 diabetes, with a 20% increased risk associated with one serving/day of potatoes [25]. We observed that individuals with a high intake of boiled potatoes had slightly higher prevalence of metabolic syndrome, defined by having adverse levels of at least three CVD risk factors. Since fasting blood glucose measurements were not obtained in the HUNT study, nor was any other measure of insulin resistance, our definition of the metabolic syndrome was based on the available data on waist circumference, blood pressure and blood lipids only. It is conceivable that people with a pre-diabetes condition such as metabolic syndrome, or with adverse levels of single CVD risk factors, could receive dietary advice that encourage them to restrict the amount of carbohydrates in the diet. This could have influenced the associations observed in our study, particularly due to the cross-sectional design. This design only allows us to describe relations between variables, and the temporal direction of effects cannot be determined. Thus, future studies should determine the longitudinal associations between intake of boiled potatoes and CVD risk factors, as well as CVD morbidity and mortality. 

## 5. Conclusions

In conclusion, we observed weak and small associations between consumption of boiled potatoes and the CVD risk factors under study. The cross-sectional design prevents us from drawing any firm conclusions.

## Figures and Tables

**Figure 1 nutrients-12-00073-f001:**
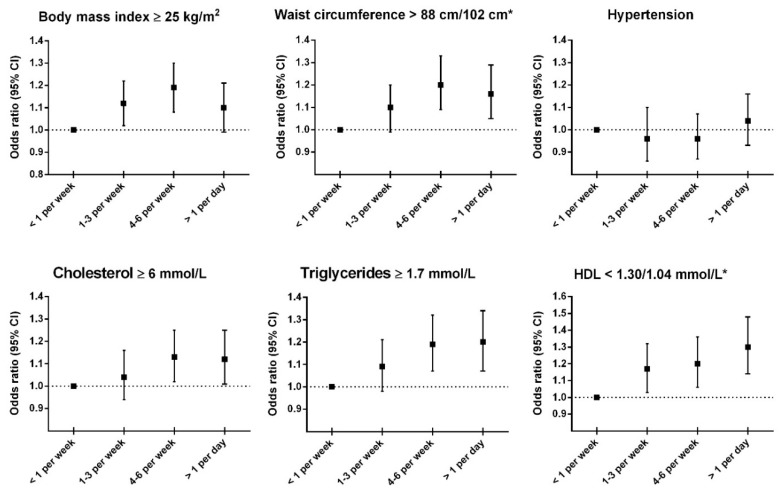
Odds ratios for adverse levels of adiposity, hypertension and blood lipids according to potato consumption. HDL = high-density lipoprotein cholesterol. * Cut-off values are sex-specific for waist circumference and HDL; Waist circumference, >88 cm for women and >102 cm for men; HDL, <1.30 mmol/L for women and <1.04 mmol/L for men. Abbreviations: HDL = high-density lipoprotein. Adjusted for age (continuous), sex (woman, man), cardiovascular disease (no, yes), work type (mostly sedentary, much walking, much lifting, heavy work, unknown/not employed), frequency of physical activity (none, <1, 1, 2–3, ≥4 times per week), smoking (never, former, current), intake of alcohol (≤1 month, ≤1 per week, ≥2 per week, abstainer), intake of vegetables (≤3 times per week, 4–6 times per week, ≥1 times per day), intake of fish (≤3 times per month, 1–3 times per week, ≥4 times per week), intake of processed meat (≤3 times per month, 1–3 times per week, ≥4 times per week), intake of pasta/rice (≤3 times per month, 1–3 times per week, ≥4 times per week). All outcomes except body mass index and waist circumference were adjusted also for body mass index.

**Table 1 nutrients-12-00073-t001:** Selected characteristics according to boiled potato consumption.

	Boiled Potato Consumption Per Week			
Variable	<1	1–3	4–6	Daily
Number of participants	2538 (6%)	11,978 (27%)	17,781 (41%)	11,386 (26%)
Age, years, mean (SD)	37.4 (14.3)	43.0 (13.1)	51.9 (13.5)	61.2 (13.0)
Women/men	1537/1001	6912/5066	9630/8151	6190/5196
Cigarette smoking, % current smoker	28%	25%	24%	26%
Alcohol consumption, % >1/week	13%	16%	17%	13%
Work type, % sedentary	32%	32%	26%	16%
Physical activity, % <1/week	21%	21%	23%	31%
Pasta/rice, % ≥4 times/week	98%	99%	98%	93%

**Table 2 nutrients-12-00073-t002:** Mean difference in components of the metabolic syndrome and cardiovascular disease risk factors according to boiled potato consumption.

Risk Factor and Frequency of Boiled Potato Consumption	Number of Persons	Mean	Crude Mean Difference	Adjusted Mean Difference ^a^	95% CI
Body mass index, kg/m^2^					
<1 per week	2538	26.2	0.0	0.0	Reference
1–3 per week	11,978	26.7	0.4	0.2	0.0 to 0.4
4–6 per week	17,781	27.2	1.0	0.4	0.2 to 0.5
≥1 per day	11,386	27.5	1.2	0.2	0.0 to 0.4
Waist circumference, cm					
<1 per week	2522	89.7	0.0	0.0	Reference
1–3 per week	11,917	91.2	1.6	0.6	0.1 to 1.0
4–6 per week	17,730	93.5	3.8	1.1	0.6 to 1.6
≥1 per day	11,366	94.9	5.3	0.9	0.3 to 1.4
SBP, mmHg					
<1 per week	2530	123.4	0.0	0.0	Reference
1–3 per week	11,937	125.4	2.1	−0.6	−1.3 to 0.9
4–6 per week	17,732	130.1	6.8	−0.6	−1.3 to 0.7
≥1 per day	11,337	135.5	12.1	0.0	−0.8 to 0.7
DBP, mmHg					
<1 per week	2530	69.8	0.0	0.0	Reference
1–3 per week	11,935	71.8	2.0	0.4	−0.1 to 0.8
4–6 per week	17,728	74.0	4.3	0.5	0.0 to 0.9
≥1 per day	11,338	74.9	5.1	−0.1	−0.6 to 0.3
Total cholesterol, mmol/L					
<1 per week	2439	5.11	0.00	0.00	Reference
1–3 per week	11,628	5.31	0.20	0.05	0.00 to 0.09
4–6 per week	17,363	5.58	0.47	0.10	0.06 to 0.15
≥1 per day	11,126	5.78	0.67	0.09	0.05 to 0.14
Triglycerides, mmol/L ^b^					
<1 per week	2490	0.22	0.00	0.00	Reference
1–3 per week	11,820	0.26	0.05	0.01	−0.01 to 0.03
4–6 per week	17,609	0.34	0.12	0.03	0.01 to 0.05
≥1 per day	11,264	0.40	0.22	0.04	0.02 to 0.06
HDL, mmol/L					
<1 per week	2439	1.36	0.00	0.00	Reference
1–3 per week	11,628	1.35	−0.02	−0.02	−0.04 to −0.01
4–6 per week	17,362	1.35	−0.01	−0.03	−0.04 to −0.02
≥1 per day	11,126	1.37	0.01	−0.03	−0.05 to −0.02

Abbreviations: SBP = systolic blood pressure; DBP = diastolic blood pressure; HDL = high-density lipoprotein; ^a^ Adjusted for age (continuous), sex (woman, man), cardiovascular disease (no, yes), work type (mostly sedentary, much walking, much lifting, heavy work, unknown/not employed), frequency of physical activity (none, <1, 1, 2–3, ≥4 times per week), smoking (never, former, current), intake of alcohol (≤1 month, ≤1 per week, ≥2 per week, abstainer), intake of vegetables (≤3 times per week, 4–6 times per week, ≥1 times per day), intake of fish (≤3 times per month, 1–3 times per week, ≥4 times per week), intake of processed meat (≤3 times per month, 1–3 times per week, ≥4 times per week), intake of pasta/rice (≤3 times per month, 1–3 times per week, ≥4 times per week) All outcomes except body mass index and waist circumference were adjusted also for body mass index. ^b^ Geometric mean from log-transformed values and corresponding ratio between geometric means.

**Table 3 nutrients-12-00073-t003:** Mean difference in cardiovascular disease risk factors according to potato consumption, stratified by sex.

	Women			Men			Inter-action
Risk Factor and Frequency of Boiled Potato Consumption	No. of Persons	Adjusted Mean Difference ^a^	95% CI	No. of Persons	Adjusted Mean Difference ^a^	95% CI	*p*
Body mass index, kg/m^2^							
<1 per week	1537	0.0	Reference	1001	0.0	Reference	<0.01
1–3 per week	6912	0.2	0.0 to 0.5	5066	0.1	−0.2 to 0.3	
4–6 per week	9630	0.4	0.2 to 0.7	8151	0.2	−0.1 to 0.4	
≥1 per day	6190	0.2	−0.1 to 0.5	5196	0.1	−0.2 to 0.3	
Waist circumference, cm							
<1 per week	1522	0.0	Reference	1000	0.0	Reference	0.01
1–3 per week	6852	0.7	0.1 to 1.4	5066	0.1	−0.6 to 0.8	
4–6 per week	9585	1.5	0.9 to 2.2	8151	0.4	−0.3 to 1.1	
≥1 per day	6178	1.0	0.2 to 1.7	5196	0.3	−0.4 to 1.1	
SBP, mmHg							
<1 per week	1531	0.0	Reference	999	0.0	Reference	<0.01
1–3 per week	6887	−0.6	−1.5 to 0.3	5050	−0.8	−1.9 to 0.2	
4–6 per week	9604	−0.7	−1.6 to 0.3	8128	−0.7	−1.8 to 0.4	
≥1 per day	6159	0.2	−0.8 to 1.2	5178	−0.5	−1.6 to 0.7	
DBP, mmHg							
<1 per week	1531	0.0	Reference	999	0.0	Reference	0.02
1–3 per week	6887	0.5	−0.1 to 1.1	5050	0.2	−0.5 to 0.9	
4–6 per week	9604	0.7	0.1 to 1.3	8126	0.3	−0.5 to 1.0	
≥1 per day	6159	0.3	−0.3 to 0.9	5177	−0.7	−1.4 to 0.1	
Total cholesterol, mmol/L							
<1 per week	1470	0.00	Reference	969	0.00	Reference	<0.01
1–3 per week	6692	0.00	−0.06 to 0.05	4936	0.11	0.04 to 0.18	
4–6 per week	9384	0.05	−0.01 to 0.10	7979	0.18	0.10 to 0.25	
≥1 per day	6036	0.03	−0.03 to 0.10	5090	0.18	0.10 to 0.26	
Triglycerides, mmol/L ^b^							
<1 per week	1503	0.00	Reference	987	0.00	Reference	<0.01
1–3 per week	6809	−0.03	−0.06 to -0.01	5011	0.06	0.03 to 0.10	
4–6 per week	9538	−0.01	−0.03 to 0.02	8071	0.08	0.05 to 0.11	
≥1 per day	6109	0.01	−0.02 to 0.03	5155	0.09	0.05 to 0.13	
HDL, mmol/L							
<1 per week	1470	0.00	Reference	969	0.00	Reference	
1–3 per week	6692	−0.03	−0.05 to −0.01	4936	−0.02	−0.04 to 0.00	0.38
4–6 per week	9383	−0.04	−0.06 to −0.02	7979	−0.02	−0.04 to 0.00	
≥1 per day	6036	−0.04	−0.06 to −0.02	5090	−0.02	−0.04 to 0.00	

Abbreviations: SBP = systolic blood pressure; DBP = diastolic blood pressure; HDL = high-density lipoprotein; ^a^ Adjusted for age (continuous), cardiovascular disease (no, yes), work type (mostly sedentary, much walking, much lifting, heavy work, unknown/not employed), frequency of physical activity (none, <1, 1, 2–3, ≥4 times per week), smoking (never, former, current), intake of alcohol (≤1 month, ≤1 per week, ≥2 per week, abstainer), intake of vegetables (≤3 times per week, 4–6 times per week, ≥1 times per day), intake of fish (≤3 times per month, 1–3 times per week, ≥4 times per week), intake of processed meat (≤3 times per month, 1–3 times per week, ≥4 times per week), intake of pasta/rice (≤3 times per month, 1–3 times per week, ≥4 times per week). All outcomes except BMI and waist circumference were adjusted also for BMI. ^b^ Geometric mean from log-transformed values and corresponding ratio between geometric means.

**Table 4 nutrients-12-00073-t004:** Mean difference in cardiovascular disease risk factors according to potato consumption, stratified by body mass index (BMI).

	<25 kg/m^2^			≥25 kg/m^2^			Inter-action
Risk Factor and Frequency of Boiled Potato Consumption	No. of Persons	Adjusted Mean Difference ^a^	95% CI	No. of Persons	Adjusted Mean Difference ^a^	95% CI	*p*
Waist circumference, cm							
<1 per week	1134	0.0	Reference	1388	0.0	Reference	<0.01
1–3 per week	4597	0.5	0.0 to 0.9	7314	−0.3	−0.8 to 0.4	
4–6 per week	5651	0.7	0.3 to 1.2	12,073	0.2	−0.4 to 0.7	
≥1 per day	3298	0.4	−0.2 to 0.9	8060	0.4	−0.2 to 1.0	
SBP, mmHg							
<1 per week	1137	0.0	Reference	1393	0.0	Reference	0.48
1–3 per week	4597	−0.5	−1.6 to 0.5	7340	−0.6	−1.6 to 0.3	
4–6 per week	5651	−0.7	−1.7 to 0.4	12,081	−0.6	−1.6 to 0.3	
≥1 per day	3298	0.3	−0.9 to 1.4	8039	−0.2	−1.2 to 0.8	
DBP, mmHg							
<1 per week	1137	0.0	Reference	1393	0.0	Reference	0.40
1–3 per week	4595	0.1	−0.6 to 0.8	7340	0.5	−0.1 to 1.1	
4–6 per week	5650	0.1	−0.6 to 0.8	12,078	0.7	0.1 to 1.3	
≥1 per day	3299	−0.4	−1.1 to 0.4	8039	0.0	−0.6 to 0.7	
Total cholesterol, mmol/L							
<1 per week	1087	0.00	Reference	1352	0.00	Reference	<0.01
1–3 per week	4476	0.00	−0.06 to 0.07	7152	0.05	−0.01 to 0.11	
4–6 per week	5524	0.04	−0.03 to 0.11	11,839	0.11	0.05 to 0.17	
≥1 per day	3260	0.02	−0.06 to 0.09	7866	0.12	0.05 to 0.18	
Triglycerides, mmol/L ^b^							
<1 per week	1114	0.00	Reference	1376	0.00	Reference	<0.01
1–3 per week	4551	−0.03	−0.06 to 0.00	7269	0.04	0.01 to 0.07	
4–6 per week	5612	0.01	−0.04 to 0.02	11,997	0.05	0.02 to 0.08	
≥1 per day	3291	0.01	−0.03 to 0.04	7973	0.06	0.03 to 0.09	
HDL, mmol/L							
<1 per week	1087	0.00	Reference	1352	0.00	Reference	
1–3 per week	4476	−0.03	−0.05 to −0.01	7152	−0.02	−0.04 to −0.01	0.05
4–6 per week	5524	−0.03	−0.06 to −0.01	11,838	−0.03	−0.04 to −0.01	
≥1 per day	3260	−0.04	−0.06 to −0.01	7866	−0.03	−0.05 to −0.01	

Abbreviations: SBP = systolic blood pressure; DBP = diastolic blood pressure; HDL = high-density lipoprotein; ^a^ Adjusted for age (continuous), sex (woman, man), cardiovascular disease (no, yes), work type (mostly sedentary, much walking, much lifting, heavy work, unknown/not employed), frequency of physical activity (none, <1, 1, 2–3, ≥4 times per week), smoking (never, former, current), intake of alcohol (≤1 month, ≤1 per week, ≥2 per week, abstainer), intake of vegetables (≤3 times per week, 4–6 times per week, ≥1 times per day), intake of fish (≤3 times per month, 1–3 times per week, ≥4 times per week), intake of processed meat (≤3 times per month, 1–3 times per week, ≥4 times per week), intake of pasta/rice (≤3 times per month, 1–3 times per week, ≥4 times per week). All outcomes except waist circumference were adjusted also for body mass index. ^b^ Geometric mean from log-transformed values and corresponding ratio between geometric means.

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
