# Peer review of "Intake of Boiled Potato in Relation to Cardiovascular Disease Risk Factors in a Large Norwegian Cohort: The HUNT Study"

_nutrients, 2019, doi:10.3390/nu12010073_

Round 1
Reviewer 1 Report
Moholt and colleagues have conducted an interesting study addressing the association between boiled potato consumption and metabolic health. Given the high consumption of potatoes, this is an interesting topic. Nevertheless, I have some comments that may be worthy to address.
Abstract: I wonder why the authors state that the associations that they have observed are "negligible". I believe this was not directly addressed int he discussion neither.
Methods section:
What was the response rate in the HUNT3 study? Which tool was used to collect information on diet? Had it been previously validated? Did the authors have information on boiled potato consumption with or without skin? Nutrient profile of boiled potato intake may be different based on this. Did the authors have information on potato consumption with preparation other than boiling? Participants who consume more boiled potatoes may be consuming also potation which ave been e.g. deep-fried. Thus, it would be important to adjust for this trait.
Results section:
Table 1: Please, provide information on all potential confounders that have been adjusted for int he multivariable analyses Table 2: please, display the tables so that they take as little pages as possible and repeat the headings on the second pages where needed For stratified analyses (e.g. Tables 3 and 4), please, show p value for interactions
Discussion section:
According to the STROBE guidelines, limitations of the study should be addressed in the discussion section. There is no limitation subsection as such in the discussion section. This study has some important limitations that would need to be addressed in the discussion section (the following items do not mean to be an exhaustive list): lack of adjustment for total caloric intake, lack of adjustment for other type of potato consumption (if not available for adjustment), cross-sectional design, validation of the tool for collecting dietary information (if not validated), etc There is no proper conclusion at the end of the discussion section. Please, add it.
References: Please, review the citation of reference 1.
Reviewer 2 Report
Moholt et al., and colleagues have investigated the association between the potato consumption on cardiovascular risk factors. Even though the study is interesting there are several drawbacks- specially statistical ones that need to be addressed before any conclusions or interpretations can be drawn from this paper. Currently mostly only association analysis has been performed. Statistical analysis to investigate Inter-individual group differences are also needed. Further clarification on the health data/family history collection on these participants is also essential. Please find some of my concerns below:
Table 1: A statistical analysis to test the difference between each group should be run and reported. For the number of participants – the % of total participants for each group should be reflected. The authors have tested the association between various clinical characteristics and the consumption of potatoes. However, no statistical tests have been reported to test the statistical difference between these groups. I feel its important for the authors to test the differences between the group to bet a better understanding of the implications of the potato consumption on the clinical characteristics Did the authors collect family history of hypertension, diabetes or CVD in these participants? A sensitivity analysis on the family history of these given their high heritability will be insightful. Was other form of potato consumption recorded in these patients – could this influence the reference group. The authors should address few aspects of hypertension and how it influences the analysis overall. A limitations paragraph can be added if needed. Were the hypertensive participants separated as those with or without medications as that might influence Author Response
Please see the attachment

Round 2
Reviewer 1 Report
I appreciate the effort taken by the authors to answer all the queries.
As for the new version, I just have some minor comment. It might be worthy to add in the discussion section if and how the 54% response rate might have affected the results.
Author Response
Dear Reviewer
Thank you for your suggestion! We have now added this sentence to the limitation section of the manuscript:
Even if we cannot exclude that the response rate of 54% affected the results, it has been argued that representativeness is not a prerequisite for generalization [23].
The new reference #23 is:
Rothman KJ, Gallacher JE, Hatch EE. Why representativeness should be avoided. International journal of epidemiology. 2013;42(4):1012-4.